# Non-Nitrogen-Fixers or Nitrogen-Fixers? Factors Distinguishing the Dominance of Chroococcal and Diazotrophic Cyanobacterial Species

**DOI:** 10.3390/ijerph192315980

**Published:** 2022-11-30

**Authors:** Elżbieta Wilk-Woźniak, Ewa Szarek-Gwiazda, Edward Walusiak, Joanna Kosiba, Wojciech Krztoń

**Affiliations:** Institute of Nature Conservation, Polish Academy of Sciences, al. Adama Mickiewicza 33, 31-120 Kraków, Poland

**Keywords:** blooms, ecology, decision tree, filamentous species, sulphate

## Abstract

Global warming and eutrophication are the main factors driving the development of cyanobacterial dominance in aquatic ecosystems. We used a model linking water temperature, oxygen saturation, concentrations of PO_4_^3−^, NO_3_^−^, NH_4_^+^, total dissolved iron (TDFe), and SO_4_^2−^ to cyanobacteria to test the turnover patterns of cyanobacterial dominance of non-nitrogen-fixing (chroococcal species) and nitrogen-fixing (filamentous diazotrophic) species. Statistical analysis was performed using decision trees. The dominance patterns of the two morphologically and ecologically distinct cyanobacterial species were associated with different environmental factors. However, SO_4_^2−^ was the most important factor that explained whether non-nitrogen-fixing or nitrogen-fixing species would dominate. Other important factors were water temperature, phosphate concentration, and oxygen saturation. The model for dominance of non-nitrogen-fixing species used SO_4_^2−^, PO_4_^3−^, and water temperature (upper layers), and SO_4_^2−^, the ratio of PO_4_^3−^/NH_4_^+^, and oxygen saturation (bottom layers). In contrast, water temperature, SO_4_^2−^, and NH_4_^+^ in the upper layers and SO_4_^2−^, NH_4_^+^, and water temperature in the bottom layers were used for the dominance of nitrogen-fixing species. The dominance of *Aphanizomenon flos-aquae* was explained by different sets of variables, indicating the presence of different strains of this species. The other cyanobacteria species showed dominance patterns that could be explained by one set of variables. As cyanobacterial blooms proliferate due to climate change, it is important to know which factors, in addition to phosphorus and nitrogen, are crucial for the mass development of the various cyanobacterial species.

## 1. Introduction

Global warming and eutrophication are well-established as factors promoting the development of cyanobacterial blooms and dominance [1,2,3,4]. The development of massive amounts of cyanobacteria leads to a decrease in water transparency, resulting in an increase in pH [5], and to the accumulation of large amounts of organic matter, the degradation of which leads to a significant depletion of the oxygen concentration in an aquatic environment [6], with lethal effects on other groups such as zooplankton, fish, and macroinvertebrates [5,7]. In addition, many cyanobacterial species produce toxins and release them into the water, making them unsuitable for drinking and recreational use. Blooms cause economic damage by affecting tourism, agriculture [8], and aquaculture industries [9]. However, cyanobacterial blooms are not an ecological endpoint, but rather an intermediary [10]; they cause changes in biodiversity and functionality of other communities [11,12], but also respond to changes in aquatic ecosystems. All these negative feedbacks make the spread and intensification of cyanobacterial blooms a problem that is increasing worldwide [3]. The most common planktonic freshwater cyanobacterial species blooms worldwide include species of the genera *Microcystis*, *Aphanizomenon*, and *Dolichospermum* [13,14,15]. *Microcystis* species are colonial and produce hepatotoxins (microcystins; [14]). Some of them, such as *Microcystis aeruginosa*, are also capable of producing a wide range of other secondary metabolites, e.g., nonribosomal peptides, cyanobactins, and microviridins, which also have the potential to disrupt aquatic ecosystems [16]. In contrast, the widely distributed filamentous diazotrophic freshwater cyanobacterial species *Aphanizomenon flos-aquae* and *Dolichospermum* spp. are capable of producing cylindrospermopsin and anatoxin-a (*Aphanizomenon flos-aquae*), and microcystins (*Dolichospermum* spp.; [14]). Filamentous genera are also the most abundant producers of taste and odour compounds in surface waters and cause a greater nuisance than *Microcystis* blooms [17]. In eutrophic waters, both chroococcal and diazotrophic filamentous cyanobacteria cause water blooms. However, due to their different morphology, physiology, and ecology, their adaptation to environmental conditions is different. As cyanobacterial blooms proliferate due to climate change, it is important to know what factors, in addition to phosphorus and nitrogen, are critical for the mass development of the various cyanobacterial species. Different species of cyanobacteria produce different types of toxins that affect other organisms and the aquatic food web in different ways, it is important to know which conditions favour the development of certain species.

Cyanobacterial growth is controlled by biotic and abiotic factors. The main macronutrients for which cyanobacteria compete are phosphorus (P) and nitrogen (N) [18]. However, some researchers extend the interaction to the N:P ratio [19,20] and also to iron and sulphate concentration [21,22]. Molot et al. [23] published a hypothetical model that related anoxia and the concentration of P, N, Fe^2+^, and sulphate to the formation of cyanobacterial blooms and formulated a hypothesis based on a hypothetical decision tree. The hypothesis determined the outcome of competition between eukaryotic phytoplankton and cyanobacteria, but did not consider the role of Fe^2+^ in competition between cyanobacteria, although they found that diazotrophic species have higher iron (Fe) requirements. Here, we examined field data on cyanobacterial dominance of chroococcales, *Microcystis ichthyoblabe* (G. Kunze) Kützing and *Woronichinia naegeliana* Unger (Elenkin), in comparison to filamentous diazotrophic heterocystous cyanobacteria, *Aphanizomenon flos-aquae* Ralfs ex Bornet and Flahault and *Dolichospermum planctonicum* (Brunnthaler) Wacklin, L. Hoffmann and Komárek. We hypothesised that the dominance of chroococcal or diazotrophic heterocytous cyanobacteria species was driven by: concentrations of PO_4_^3−^, NO_3_^−^, NH_4_^+^, total dissolved iron (TDFe) and SO_4_^2−^ concentrations, oxygen saturation and water temperature, but as different models. The aim of our studies was to explain the outcome of chroococcal or diazotrophic cyanobacteria domination.

## 2. Materials and Methods

Samples were collected from May to October in four shallow, eutrophic waters (max depth of 4 m; Table 1) in southern Poland (near Kraków, Table 1). In the months when cyanobacterial blooms did not occur, samples were collected once a month and every week after the beginning of the bloom. Because the development and dominance of cyanobacteria is a consequence of factors present in the upper layer and factors and processes occurring in the bottom layer, we collected samples, for determination of physical and chemical parameters, from the upper (1 m) and from the bottom water layer (approximately 10 cm above bottom sediments). A total of 62 samples were collected for physical, chemical, and biological analyses. Water temperature, oxygen saturation and chlorophyll *a* were measured in situ using a YSI 6600 V2 multiparameter probe. Samples for ion analysis (SO_4_^2−^, NO_3_^−^, PO_4_^3−^, NH_4_^+^) were taken immediately to the laboratory. Ion concentrations were measured using a Dionex ion chromatograph (DIONEX, IC25 ion chromatograph; ICS-1000, Sunnyvale, CA, USA) in the laboratory of the Institute of Nature Conservation, Polish Academy of Sciences. Water samples for total dissolved iron (TDFe) analysis were filtered through a syringe filters with a pore size of 0.45 µm into a polyethylene container. The filtered solution was acidified to pH 2 with ultrapure HNO_3_. The concentrations of dissolved Fe were measured using the atomic absorption spectrophotometric AAS method on a Varian spectrophotometer (Spectr AA-20). Following [24], we used TDFe as a proxy Fe^2+^ since Fe^2+^ is very difficult to measure in the field. Samples for cyanobacteria analysis were collected from 1 m depth using a 5 L Ruttner sampler and concentrated from 10 L using a plankton net (mesh size 10 μm). Immediately after collection, samples were fixed with Lugol’s solution for quantitative analyses. Additional fresh (non-fixed) samples were concentrated as described above, and live material was collected for qualitative analysis (species composition), which was performed immediately in the laboratory under the light microscope. Cyanobacterial species were identified and counted in a modified chamber (0.4 mm high, 22 mm diameter). We used the keys [25,26,27] to identify the cyanobacteria species. The biomass of cyanobacteria was calculated based on cell numbers and specific geometric figures [28]. A NikonH550 L light microscope at 40–1000× was used for the analyses. 

Trophic state index (TSI) was calculated based on chlorophyll *a* concentration: TSI (CHL, μg L^−1^) = 10 × [6 − (2.04 − 0.68 ln (CHL-a))/ln2] [29].

**Table 1 ijerph-19-15980-t001:** Basic information on water bodies. Mean TSI values calculated for chlorophyll a. Thresholds for TSI according to [30]: oligotrophy < 40, mesotrophy 40–50, eutrophy 50–70, hypertrophic > 70.

	Type of Reservoir	Geographical Coordinates	Supply by River	Max Depth [m]	Surface [ha]	TSI Index	Dominated Cyanobacteria
Piekary	Oxbow lake	50°00′50.1′′ N, 19°47′35.7′′ E	Vistula	4	1.6	64.7 eutrophy	*Dolichospermum* spp.
Tyniec	Oxbow lake	50°01′47′′ N, 19°49′39.8′′ E	Vistula	3	5.75	66.1 eutrophy	*Microcystis ichthyoblabe* and *Woronichinia naegeliana*
Podkamycze 1	Artificial pond	50°05′11′′ N, 19°50′01.6′′ E	Rudawa	3	16.82	57.8 eutrophy	*Aphanizomenon flos-aqaue*
Podkamycze 2	Artificial pond	50°04′59.6′′ N, 19°50′05.4′′ E	Rudawa	2	17.28	65.1 eutrophy	*Aphanizomenon flos-aqaue*

### Statistical Analyses

Differences in water temperature and oxygen saturation between water layers were tested using generalized linear models (GLMs) with a categorical predictor. The significance level was set at *p* < 0.05. To determine which factors control cyanobacterial development and dominance, we used decision trees (package ‘rpart’). This method is commonly used to test a categorical dependent variable with categorical and numerical variables. Moreover, the results obtained from the analysis are easy to interpret. Each of the nodes of the final tree has a set of values of the predictors used, which determine the exact conditions that promoted the specific category of the dependent variable. Finally, the decision tree approach does not require any additional assumptions, making the method applicable in numerous domains. The depth of each decision tree was manually determined to avoid overfitting the prediction. Data were analyzed using the statistical software R and R Studio [31].

## 3. Results

The physico-chemical parameters of the studied waters are shown in Table 2.

Despite the fact that all waters were relatively shallow, we found that oxygen saturation was significantly higher in the upper water layers than in the bottom layers (*p* = 0.0185, GLM). Water temperatures in the upper layers were also higher than those in the bottom layers, although the analysis showed a result (GLM) on the border of significance (*p* = 0.0636).

Cyanobacteria were present in all studied waters and dominated the phytoplankton community in summer and fall. Among nitrogen-fixing (filamentous diazotrophic) species, the most abundant with the highest biomass was *Aphanizomenon flos-aquae* and *Dolichospermum planctonicum*; among non-nitrogen-fixing (chroococcales) species, *Microcystis ichthyoblabe* and *Woronichinia naegeliana* were the most abundant.

### 3.1. Upper Layer-Decision Tree

Factors examined in the upper layers explained why 74% of the samples had a dominant cyanobacterial species and 26% of the samples did not have a dominant cyanobacterial species (Figure 1). Non-nitrogen-fixing (chroococcales) species of cyanobacteria dominated 17% of the samples; 10% of the samples were dominated by *M. ichthyoblabe* and 7% of the samples were dominated by *W. naegeliana*. Nitrogen-fixing (filamentous, diazotrophic) species dominated 57% of the samples; 49% were dominated by *A. flos-aquae* and 8% by *D. planctonicum*.

The most important factor that distinguished the dominance of non-nitrogen-fixing (chroococcales) from the dominance of nitrogen-fixing (filamentous diazotrophic) cyanobacteria in the upper layers was the sulphate concentration (Figure 1). When the SO_4_^2−^ concentration was higher than 69 mg/L, the non-nitrogen-fixing cyanobacteria dominated. When the PO_4_^3−^ concentration was higher than 0.029 mg/L and the water temperature was above 16 °C, *M. ichthyoblabe* dominated (10% of the samples). However, when the PO_4_^3−^ concentration was higher than 0.029 mg/L and the water temperature was below 16 °C, *W. naegeliana* dominated (7% of samples).

The circumstances for the dominance of nitrogen-fixing species were more complex. All samples dominated by *A. flos-aquae* and *D. planctonicum* were found at SO_4_^2−^ concentrations less than 69 mg/L. However, *A. flos-aquae* dominated when SO_4_^2−^ concentration was between 69 and 39 mg/L and the water temperature was above 16 °C (44% of samples). When the water temperature was above 16 °C but SO_4_^2−^ concentration was below 39 mg/L, *D. planctonicum* dominated (8% of the samples).

We also found that the samples were dominated by *A. flos-aquae* (5%) when the SO_4_^2−^ concentration was below 41 mg/L, the water temperature was below 16 °C, and the NH_4_^+^ concentration was below 0.11 mg/L. We found that the PO_4_^3−^:NH_4_^+^ ratio and dissolved iron (TDFe) and NH_4_^+^ concentrations were less important for the dominance of cyanobacteria in the upper layers (description of Figure 1).

### 3.2. Decision Tree- Near-Bottom Layer

The factors present in the bottom layers explained why 76% of the samples had a dominant cyanobacterial species and 24% of the samples had no dominant cyanobacterial species (Figure 2).

Overall, seventeen percent of the samples were dominated by chroococcal cyanobacteria, and 59% of the samples were dominated by nitrogen-fixing species. In the bottom layers, SO_4_^2−^ was also the main factor that distinguished the samples dominated by non-nitrogen-fixing species from those dominated by nitrogen-fixing cyanobacteria. For *Microcystis* spp. dominance, the most important factors were a SO_4_^2−^ concentration greater than 73 mg/L, a PO_4_^3−^:NH_4_^+^ ratio greater than 0.051, and an oxygen saturation greater than 7.5%. These conditions occurred in 12% of the samples. In the case of *W. naegeliana*, the factor that distinguished the dominance of *W. naegeliana* from the dominance of *M. ichthyoblabe* was an oxygen saturation of less than 7.5%. 

The dominance of nitrogen-fixing cyanobacteria was observed under conditions with near-bottom SO_4_^2−^ concentrations of less than 73 mg/L. We found that 47% of samples dominated by *A. flos-aquae* occurred at near-bottom conditions of SO_4_^2−^ concentration between 53–73 mg/L. When the SO_4_^2−^ concentration was below 53 mg/L and the NH_4_^+^ concentration was above 0.37 mg/L, *D. planctonicum* predominated (9% of the samples). Three percent of the samples were dominated by *A. flos-aquae* when SO_4_^2−^ concentration was below 53 mg/L, NH_4_^+^ concentration was between 0.18–0.37 mg/L, and the water temperature was above 14 °C. We found that PO_4_^3−^ and total dissolved iron (TDFe) were less important factors (description of Figure 2).

## 4. Discussion

Two morphologically and ecologically distinct species of cyanobacteria, non-nitrogen-fixing (chroococcales) and nitrogen-fixing (filamentous diazotrophs), dominated the phytoplankton in four shallow small waters. The dominance patterns of each species were associated with various environmental factors. However, the decision trees showed that SO_4_^2−^, in both the upper and bottom layers, was the most important factor explaining whether non-nitrogen-fixing (chroococcales) or nitrogen-fixing (diazotrophs) species dominated. This finding suggests that sulphate concentration in water should be considered when preparing the management of cyanobacterial dominance and blooms. In the upper and bottom layers, sulphate separated the non-nitrogen fixers from the nitrogen fixers. Higher SO_4_^2−^ concentrations were associated with non-nitrogen-fixing species (17% of samples) and lower SO_4_^2−^ was associated with nitrogen-fixing species (57% of samples). The effect of sulphate concentration on cyanobacterial dominance can be explained in several ways. Sulphate is an important inhibitor of molybdate uptake in natural waters, and molybdenum (Mo) is one of the essential cofactors for the vast majority of known N_2_ fixation systems and many nitrate reductases [21]. Some authors point out that SO_4_^2−^ concentrations, which are typically 4–6 times higher than molybdate concentrations, can inhibit molybdate uptake [21]. The Mo demand depends on what form of nitrogen is used by the specimen. For example, molybdenum (Mo) demand is the highest when organisms fix N_2_, lower (but still present) when organisms are supplied with NO_3_^−^, and negligible when organisms are supplied with NH_4_^+^ [32]. The low availability of Mo caused by a high concentration of SO_4_^2−^ in oxic waters may limit the activity of planktonic N_2_-fixing organisms [21]. At high concentrations of SO_4_^2−^ in water, nitrogen fixation may be difficult or even impossible. In nitrogen-fixing species such as *A. flos-aquae*, inhibition of Mo would block nitrogen fixation, while in *Microcystis*, inhibition of Mo should not cause problems for nitrogen uptake [33]. However, there are some studies showing that nitrogen-fixing cyanobacteria were not limited by Mo in a group of eutrophic saline lakes with high sulphate/Mo ratios [34], so the above explanation may be questionable. 

Sulphates can also limit N_2_ fixation [35]. A high SO_4_^2−^ concentration favours the development of certain species, i.e., either those that take up nitrogen or those that fix nitrogen. Our results show that a lower sulphate concentration favours the dominance of diazotrophic species and a higher sulphate concentration favours the dominance of chroococcal type cyanobacteria. Although filamentous diazotrophic cyanobacteria are able to fix N_2_ [19], we cannot ignore that both types of cyanobacteria, chroococcales and diazotrophs, are able to use NO_3_^−^, NH_4_^+^ and urea as nitrogen sources [33]. Because we did not measure nutrient uptake or nitrogen fixation rates, the discussion of our results remains partially hypothetical, but the idea is worth developing in the future. It is also worth developing field studies that focus on the presence/absence and abundance (ratio of heterocytes to vegetative cells) of heterocytes as a function of sulphate concentration and different types of nitrogen sources. Other important factors separating the different cyanobacterial species were water temperature, phosphate concentration, ammonia nitrogen, and oxygen saturation. Phosphate concentration was important for the dominance of non-fixers (chroococcales). For example, the abundance of *Microcystis* was shown to be phosphorus responsive and required a large amount of phosphorus that could be rapidly taken up [33,36]. It has also been demonstrated that the genus *Microcystis* has a lower tolerance to phosphorus (P) stress than two other filamentous species, *Aphanizomenon flos-aquae* and *Oscillatoria planctonica* [37], and that the biomass of *Microcystis* is affected by phosphorus concentration [38]. In our studies, *Microcystis* dominated in samples with high PO_4_^3−^ and SO_4_^2−^ concentrations. The source of phosphates in the studied waters could also be internal loading. The presence of weakly reducing conditions promotes the release of phosphates from the redox-sensitive iron-bound P fraction in the sediment into the overlying water. This phenomenon is influenced by redox potential values below 200 mV, which are typical of dissolved oxygen content 2–4 dm^−3^ in the bottom water [39]. The iron-bound P fraction can contain up to 78% of the total P content in the sediment [40,41]. SO_4_^2−^ has an important function in controlling the release of Fe^2+^ and PO_4_^3−^ from sediments during anoxia [42]. Sulphate reduction limits Fe^2+^ availability through the formation of iron sulphide, which is consistent with the Fe^2+^ availability hypothesis [23]. Such a process could have occurred in the oxbow lakes of Tyniec and Piekary at a low oxygen saturation of bottom waters. 

Two species of dominant chroococcal cyanobacteria have been observed: *Microcystis* sp. and *Woronichinia naegeliana*. The factor that distinguished the spread of *Microcystis* from that of *Woronichinia* in the upper layers was water temperature, while the factor in the bottom layers was oxygen saturation. *Woronichinia* preferred temperatures lower than 16 °C and tolerated oxygen saturation lower than 7.5%. The preference for low temperatures fits the phenomenon of the autumnal bloom of *W. naegeliana* in deep submontane reservoirs [43]. The ability of *W. naegeliana* to tolerate low oxygen saturation has also been noted in several lakes [44]. This characteristic could be useful for nutrient uptake. Under reducing conditions in the water-sediment system of the studied waters, NH_4_^+^ may be released from sediments as a result of organic matter decay. In addition, under anoxic conditions, biological nitrification in the bottom layers is lost [45]. Anoxic release rates of ammonia for eutrophic sites have been found to exceed 15 mg Nm^−2^ day^−1^. It has also been found that anoxic conditions in eutrophic lakes can promote the co-occurrence of phosphorus and ammonia [45]. Under anoxic conditions, fluxes of the biologically available nutrients soluble reactive phosphate (SRP) and ammonium nitrogen from internal releases can exceed those from external sources [46]. Under such conditions, anoxia-tolerant *W. naegeliana* can utilise phosphorus (P) and nitrogen (N) released from sediments more rapidly than other species that do not tolerate anoxia. Previous studies have found a significant negative correlation between *W. naegeliana* and nitrate nitrogen [47], suggesting that this species takes up ammonia nitrogen.

The dominance of nitrogen-fixing cyanobacteria was identified as somewhat more complex compared to non-nitrogen-fixing species. Besides the main factor SO_4_^2−^ mentioned above, the second most important parameter in the upper layers was a water temperature higher than 16 °C. Experiments with cultures have shown that water temperature is crucial for the growth of *A. flos-aquae* and that growth does not occur at a temperature of 11 °C or below [48]. Wu et al. [49] showed that *A. flos-aquae* grew at different temperatures (10, 15, 20, and 25 °C) but preferred higher temperatures (about 20–25 °C). However, later [36] observed that *A. flos-aquae* grew rapidly and forms blooms at a water temperature of about 15 °C. All these observations indicate that a water temperature of about 15 °C is important for the development and dominance of *A. flos-aquae*. Temperature affects the formation and abundance of heterocysts in both *Dolichospermum* and *Aphanizomenon* [50] and is important for phosphorus and ammonia concentrations. Experimental studies conducted at 16 °C have shown that phosphorus can be released from sediments under aerobic and anaerobic conditions, and the processes were more intense under anaerobic conditions [51]. Therefore, the threshold of 16 °C for *A. flos-aquae* and *D. planctonicum* could be related to the release of phosphorus or to the ammonia concentration, which was significant in the 5% of samples dominated by *A. flos-aquae*. The ammonia concentration was lower than 0.11 mg/L; however, this concentration was sufficient. Ammonia and nitrate nitrogen are important for diazotrophic cyanobacteria because both can suppress nitrogenase [52,53]. Diazotrophic cyanobacteria often preferentially take up ammonia when it is available because it is much more energetically favourable than fixing nitrogen [52]. The type of nitrogen source is related to the availability of phosphorus and the N:P ratio [52].

In the model [23], the limitation of cyanobacterial growth by iron was demonstrated. It is known that iron (Fe^2+^) regulates the efficiency of macronutrient use by cyanobacteria, as it plays a critical role in the uptake of nitrogen (N) and phosphorus (P) [23,54]. Since Fe^2+^ is very difficult to measure in the field, we used total dissolved Fe (TDFe) as a proxy of Fe^2+^ in anoxic waters [22,24]. Although all the studied waters are shallow, we found statistically significant differences between the upper and bottom layers, indicating weaker oxygenation of bottom waters. Poor oxygenation and high total dissolved iron (TDFe) concentration in bottom waters, especially in two water bodies (Piekary, Tyniec), indicated the release of Fe^2+^ from sediment and its possible bioavailability to cyanobacteria. However, in our studies, total dissolved Fe concentration showed little significance and was not responsible for differentiating the dominance of cyanobacterial species. 

We also found that the phosphate/ammonia nitrogen ratio in soil samples differentiated between dominance of chroococcal species and a lack of dominance of cyanobacteria. A high ratio of PO_4_^3−^ to NH_4_^+^ favoured *Microcystis* or *Woronichinia* dominance (17% of samples overall), while a lower ratio favoured samples (only 5%) without cyanobacterial dominance. We hypothesise that this is related to the high phosphate requirements of the chroococcal species. Another interesting finding of our study was a small number of samples dominated by *A. flos-aquae* (3% of samples) that were favoured by other factors than the majority of *Aphanizomenon* samples (47% of samples). This may be due to our assumption that different strains of *A. flos-aquae* were present in these samples. In most waters, each bloom-forming genus actually includes multiple strains or subspecies [17] and it has been noted that the responses of *Aphanizomenon* and *Dolichospermum* to environmental conditions exhibit lake-specific patterns [38]. We suggest that the pattern we observed is due to the rate of evolution of cyanobacteria. Chroococcales are a more conservative species, and their speciation and evolution are much slower than that of the filamentous *A. flos-aquae*. This could be supported by the fact that chroococcal species appeared before filamentous species in evolution. Presumably, the first cyanobacteria were unicellular, coccoid, or had short-rod morphologies. Filamentous morphologies with the ability to fix nitrogen evolved independently several times thereafter [55].

The results of our studies are important for preventing cyanobacterial dominance. Although various methods of remediation of water bodies have been proposed to avoid the harmful consequences of cyanobacterial development [56,57], prevention is much more effective than any remedy. Therefore, knowledge of the factors that favour the development of certain cyanobacterial species could be useful in preventing the development of particularly dangerous species.

## 5. Conclusions

The results of our field study on the dominance patterns of cyanobacteria showed that SO_4_^2−^ in the upper and bottom layers was the most important factor explaining the dominance of non-nitrogen-fixing or of nitrogen-fixing species. A lower sulphate concentration favoured the dominance of nitrogen-fixing species, while a higher concentration favoured the dominance of non-nitrogen-fixing species. Sulphates could be a factor that blocks nitrogen fixation and then causes the decline of nitrogen-fixing species when they do not take up other nitrogen sources. Sulphate does not affect nitrogen uptake and causes an increase in non-nitrogen-fixing species. Sulphate appears to be the primary factor in turning on or off the dominance of various cyanobacterial species. In addition, the dominance patterns of the two morphologically and ecologically distinct cyanobacterial species were associated with various environmental factors such as phosphate, ammonium nitrogen, water temperature, oxygen saturation, and phosphate/ammonium nitrogen ratio.

## Figures and Tables

**Figure 1 ijerph-19-15980-f001:**
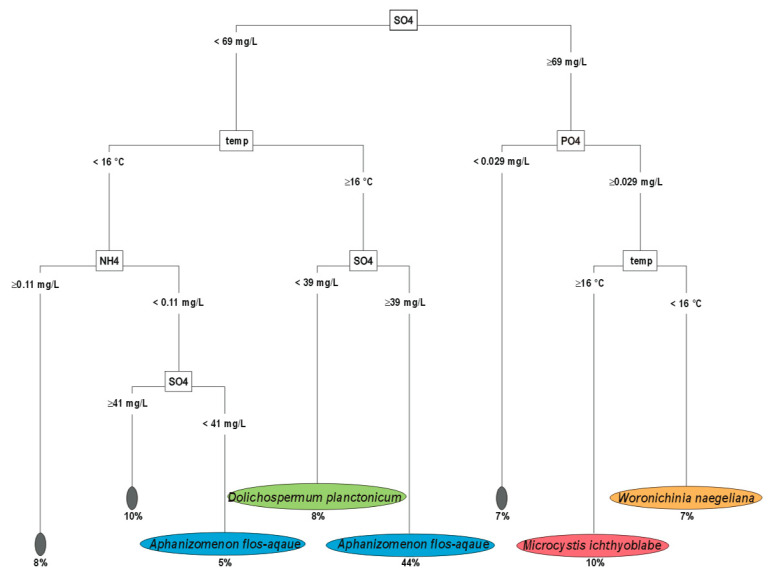
Decision tree of factors explaining the dominance of cyanobacterial species—upper layer. Small grey boxes—no dominant cyanobacteria; blue and green boxes—dominance of filamentous diazotrophic cyanobacteria (*A. flos-aquae*, *D. planctonicum*); red and orange boxes—dominance of Chroococcales (*M. ichthyoblabe*, *W. naegeliana*). Abbreviations: SO4—SO_4_^2−^ [mg/L], PO4—PO_4_^3−^ [mg/L], temp—water temperature °C], NH4—NH_4_^+^ [mg/L]. The importance of each factor was as follows: SO_4_^2−^ > water temperature > PO_4_^3−^ > PO_4_^3−^:NH_4_^+^ > total dissolved Fe (TDFe) > NH_4_^+^ (18.1 > 10.6 > 8.5 > 6.3 > 5.2 > 4.8; respectively).

**Figure 2 ijerph-19-15980-f002:**
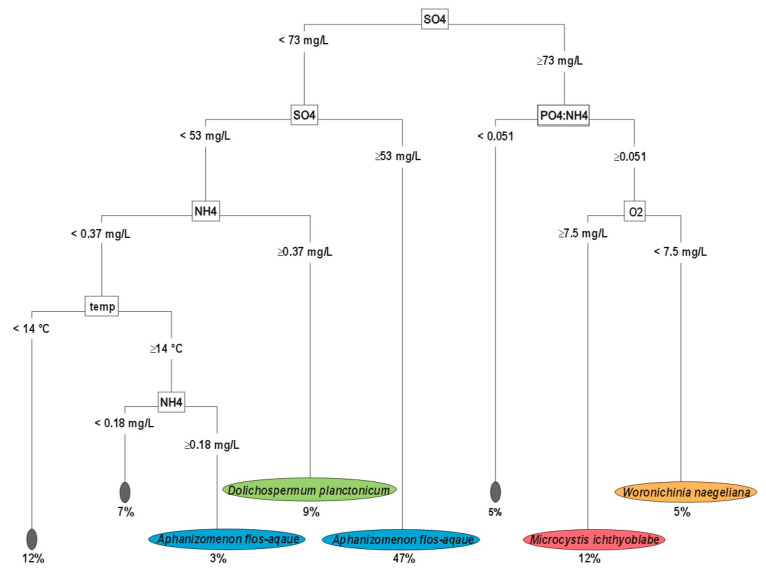
Decision tree of factors explaining the dominance of cyanobacterial species—bottom layer. Grey boxes—no dominant cyanobacteria; blue and green boxes—dominance of filamentous diazotrophic cyanobacteria (*A. flos-aquae*, *D. planctonicum*); red and orange boxes—dominance of Chroococcales (*M. ichthyoblabe*, *W. naegeliana*). Abbreviations: SO4—SO_4_^2−^ [mg/L], PO4:NH4PO_4_^3−^:NH_4_^+^ ratio, O2—oxygen saturation [%], temp—water temperature [°C], NH4—NH_4_^+^ [mg/L]. The importance of each factor was as follows: SO_4_^2−^ > NH_4_^+^ > PO_4_^3−^:NH_4_^+^ > O_2_% > water temperature > PO_4_^3−^ > total dissolved Fe (TDFe; 18.3 > 9.6 > 9.1 > 8.6 > 3.9 > 3.8, respectively).

**Table 2 ijerph-19-15980-t002:** Physico-chemical parameters of waters of the upper and lower layers of the oxbow lakes Piekary (P) and Tyniec (T) and the ponds Podkamycze 1 (P1) and Podkamycze 2 (P2).

Parameter	Layer	Water Body											
		P	T	P1	P2	P	T	P1	P2	P	T	P1	P2
		Min–Max				Average				SD			
Water temperature [°C]	Upper	8.7–24.3	9.3–24.7	7.2–23.4	8.4–25.6	17.4	17.8	17.4	18.9	4.7	4.6	4.2	4.3
	Lower	8.6–17.2	9.3–23.2	7.0–19.4	8.4–24.1	13.8	16.0	16.1	18.2	2.6	3.9	3.4	4.0
Oxygen saturation [%]	Upper	53.1–100.8	41.0–169.6	88.5–214.7	11.8–236.6	74.4	88.3	141.6	133.1	17.6	37.8	34.1	49.1
	Lower	2.6–53.3	2.2–61.2	30.3–182.1	16.4–226.2	30.7	27.4	113.4	123.5	17.2	19.5	36.5	49.2
pH	Upper	6.4–8.3	6.8–8.3	7.2–8.9	7.4–8.6	7.2	7.4	7.9	8.0	0.5	0.5	0.4	0.3
	Lower	5.5–7.5	6.4–7.7	7.5–8.6	7.6–8.5	6.5	7.0	8.0	8.0	0.6	0.4	0.3	0.3
NO_3_^−^ [mg/L]	Upper	0.18–1.03	nd-1.05	6.53–12.11	0.67–5.62	0.39	0.53	9.37	2.96	0.24	0.24	1.63	1.64
	Lower	0.14–3.50	nd-4.08	7.57–13.65	0.85–6.05	0.56	0.74	10.72	3.04	0.98	1.03	1.24	1.76
NH_4_^+^ [mg/L]	Upper	0.03–0.56	0.03–0.78	0.02–0.56	0.02–0.56	0.18	0.22	0.14	0.15	0.16	0.22	0.15	0.17
	Lower	0.11–1.96	0.06–1.21	0.02–0.43	0.01–0.48	0.58	0.28	0.12	0.12	0.55	0.31	0.11	0.14
PO_4_^3−^ [mg/L]	Upper	0–0.189	0–0.490	0–0.538	0.021–0.432	0.059	0.147	0.184	0.098	0.054	0.159	0.167	0.095
	Lower	0–0.171	0–0.517	0–0.573	0–0.432	0.060	0.117	0.230	0.082	0.048	0.146	0.172	0.103
SO_4_^2−^ [mg/L]	Upper	21.2–78.1	75.9–100.1	50.5–58.9	50.1–62.9	36.7	84.7	54.9	55.6	14.5	6.4	2.4	4.1
	Lower	17.7–63.2	79.3–91.8	52.3–58.0	50.1–66.7	33.3	84.8	54.4	55.2	11.0	3.9	1.7	4.5
Fe dissolved [µg/L]	Upper	5.6–101.0	3.4–160.0	0.8–48.0	1.2–45.0	29.3	55.5	21.8	22.1	25.9	50.0	13.3	13.5
	Lower	4.7–159.0	1.9–204.4	1.6–66.0	3.0–55.0	40.0	49.2	23.5	24.9	41.8	53.3	17.2	15.0
Cyanobacterial biomass [mg/L]		0–0.354	0–12.830	0.11–5.61	0.06–9.23	0.11	4.65	1.33	2.15	0.12	3.77	1.41	3.10

## Data Availability

Data are available from the first author after request.

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
