# Peer review of "Non-Nitrogen-Fixers or Nitrogen-Fixers? Factors Distinguishing the Dominance of Chroococcal and Diazotrophic Cyanobacterial Species"

_ijerph, 2022, doi:10.3390/ijerph192315980_

Round 1
Reviewer 1 Report
This study focused on the effects of different parameters on the dominance of non-nitrogen-fixing and nitrogen-fixing cyanobacteria species, and found that SO42- is the most important factor. The manuscript was well written, but there are still some sentences need to be improved.
1. The significance or novelty of this study should be improved in both the abstract and the introduction.
2. Line 51-52 This sentences is confused and hard to understand.
3. Table 2 The unit of Fe dissolved in the table is "g/L" and it is too high for natural water, please check it. Besides, the data of cyanobacterial biomass of lower layer is missing.
4. Line 221 "the highest"
5. Line 226-227 The author can only use the abbreviation Mo here.
6. The discussion of the effect of SO42- is a bit confusing and need to be reorganized to clarify the relevant mechanisms.
7. Some format errors need to be corrected in line 95, 113, 134, etc.
Author Response
Dear Reviewer,
thank you very much for your comments and suggestions. We have responded to all of them. They improved our manuscript deeply. Here are our replies:
- The significance or novelty of this study should be improved in both the abstract and the introduction. Re: thank you for your comments. We added the significance of our studies: “As cyanobacterial blooms proliferate due to climate change, it is important to know which factors, in addition to phosphorus and nitrogen, are crucial for the mass development of the various cyanobacterial species”.
- Line 51-52 This sentences is confused and hard to understand. Re: thank you for your comments. We have changed the statement into: “In eutrophic waters, both chroococcal and diazotrophic filamentous cyanobacteria cause water blooms.”
- Table 2 The unit of Fe dissolved in the table is "g/L" and it is too high for natural water, please check it. Besides, the data of cyanobacterial biomass of lower layer is missing. Re: We are very sorry for this error. It should be µg/l. We have changed. The biomass of cyanobacteria was collected only from the 1 m layer. We were interested in the predominance of cyanobacteria in phytoplankton. It is known that cyanobacteria float to the surface during the day and the greatest amount occurs near the surface. Therefore, we decided to use the values from the 1m layer for the analyses. Also, our previous preliminary studies have shown that the samples from the 1m layer are the most representative.
- Line 221 "the highest" Re: thank you. We have changed.
- Line 226-227 The author can only use the abbreviation Mo here. Re: Thank you. We have corrected
- The discussion of the effect of SO42- is a bit confusing and need to be reorganized to clarify the relevant mechanisms. Re: thank you. We have corrected.
- Some format errors need to be corrected in line 95, 113, 134, etc. Re: thank you. We have corrected.
Hope, that our corrections improved the manuscript and all our changes, and explanations are accepted.
Best regards,
on behalf of all authors - Elżbieta Wilk-Woźniak
Reviewer 2 Report
1. Include formula for calculating Trophic state index and its classification 2. Support with bibliographical reference the affirmation of lines 123-124 3. Describe in greater detail the technique for the determination of Cyanobacterial 4. Indicate how the values ​​are proposed for each of the ions in figures 1 and 2 5. Indicate the sulfate concentrations at which N2 fixation is inhibited 6. Check punctuation marks 7. Improve the quality and format of figures 1 and 2 8. Why are the ion concentration values ​​different in Figures 1 and 2? 9. Include bacterial count table 10. Include the results of the physicochemical parameters or the correlation of parameters with the bacterial count 11. In lines 270-271, it is mentioned that there are reducing conditions but they do not show ORP or NO3- values.Author Response
Dear Reviewer,
we are deeply thankful to you for all comments, questions and suggestions which improved our manuscript. We hope that corrections and explanations provided by us are enough good for acceptance of the manuscript. Below you will find the replies:
- Include formula for calculating Trophic state index and its classification. Re: we explained classification of Trophic index in the description of Tab. 1. Trophic index was calculated according to Carlson, R. E. A trophic state index for lakes 1. Limnology and Oceanography 1977, 22(2), 361-369, where is formula. We do not think it is necessary to rewrite the formula here.
- Support with bibliographical reference the affirmation of lines 123-124. Re: we have changed the sentence. TSI index is shown in tab.1. The previous sentence was unclear. Thank you for poiting it out.
- Describe in greater detail the technique for the determination of Cyanobacterial. Re: we made correction.
- Indicate how the values ​​are proposed for each of the ions in figures 1 and 2. Re: The values were determined by decision trees which are based on the powerful algorithms allowing analysis of complex datasets of qualitative and/or quantitative variables. The algorithms are partitioning the dataset by splitting it into nodes, basing on number of variables and their values.
- Indicate the sulfate concentrations at which N2 fixation is inhibited. Re: we do not provideddirect numbers but sulfates might inhibit N2 fixation indirect. Sulfate inhibits molybdate assimilation by cyanobacteria, and molybdenum has been considered essential components of nitrogenase required for nitrogen fixation. However, the concentration of molybdenum to nitrogen fixation in lakes vary.
- Check punctuation marks. Re: we did it.
- Improve the quality and format of figures 1 and 2. Re: we improved.
- Why are the ion concentration values ​​different in Figures 1 and 2? Re: Decision trees were applied to two datasets of physio-chemical parameters at 1 meter depth and at the bottom of each reservoir. Each of decision trees we present among the manuscript addresses mentioned sampling depths (1 meter and bottom). Since the physio-chemical were different between depths also values presented in Figures 1 and 2 are differing.
- Include bacterial count table. Re: we included cyanobacteria biomass in the Tab. 1.
- Include the results of the physicochemical parameters or the correlation of parameters with the bacterial count. Re: results of physico-chemical nalyses are included in the tabs 1 and 2.
- In lines 270-271, it is mentioned that there are reducing conditions but they do not show ORP or NO3- values. Re: this is part of discussion, and this statement is a citation of another paper. This is not our results. Our results e.g. NO3 has been shown in tab. 2.
Thank you for all your work and great help.
Best regards,
on behalf of all authors
Elżbieta Wilk-Woźniak
Round 2
Reviewer 2 Report
1. No change is observed in the figures 1 and 2.
2. Carlson, in his article "trophic state index for lakes", uses total phosphorus and not phosphates to calculate TSI. The techniques differ because one is with digestion and the other is not.
3. Cite in Table 1 the reference to the TSI classification, since Carlson's article does not refer to this classification.
Author Response
Dear Reviewer,
we are deeply thankful for your comments. We are very sorry for all mistakes which have been done. We did further correction.
Here are our replies:
- No change is observed in the figures 1 and 2. – Re: we changed fig 1 and 2 according to the instructions for authors: ‘File for Figures and Schemes must be provided during submission in a single zip archive and at a sufficiently high resolution (minimum 1000 pixels width/height, or a resolution of 300 dpi or higher). Common formats are accepted, however, TIFF, JPEG, EPS and PDF are preferred’. So, the numbers were changed as follow: TIFF, 300 dpi resolution and 1000/1300 pixels.
- Carlson, in his article "trophic state index for lakes", uses total phosphorus and not phosphates to calculate TSI. The techniques differ because one is with digestion and the other is not. – Re: thank you very much for your comment. This is indeed an important difference. To calculate TSI, we used chlorophyll a, what is mentioned in the last sentence before Table 1. We added the formula for the TSI count: TSI (CHL, μg L−1) = 10 × [6 − (2.04 − 0.68 ln (CHL-a))/ln2]
- Cite in Table 1 the reference to the TSI classification, since Carlson's article does not refer to this classification. – Re: Thank you for the comment. We have added the reference: Carlson, R. E.; Simpson, J. (1996). A coordinator’s guide to volunteer lake monitoring methods. North American Lake Management Society 1996, 96, pp. 305.
Hope you accept the changes. Thank you for your great contribution into the mansucript, which is much better now.
On behalf of all authors,
Elżbieta Wilk-Woźniak
